# Optimization of Constitutive Promoters Using a Promoter-Trapping Vector in *Burkholderia pyrrocinia* JK-SH007

**DOI:** 10.3390/ijms24119419

**Published:** 2023-05-29

**Authors:** Xue-Lian Wu, Xiao-Wei Liu, Yang Wang, Meng-Yun Guo, Jian-Ren Ye

**Affiliations:** 1Co-Innovation Center for Sustainable Forestry in Southern China, College of Forestry, Nanjing Forestry University, Nanjing 210037, China; xuelianwuletter@163.com (X.-L.W.); njfulxw@163.com (X.-W.L.); 2Jiangsu Key Laboratory for Prevention and Management of Invasive Species, Nanjing Forestry University, Nanjing 210037, China; 3Institute of Forest Pest Control, Jiangxi Academy of Forestry, Nanchang 330032, China; wy21sj@163.com; 4Key Laboratory for Bio-Resources and Eco-Environment of Ministry of Education, College of Life Science, Sichuan University, Chengdu 610065, China; mengyun_guo@163.com

**Keywords:** *Burkholderia*, promoter-trapping vector, *TP*
^r^, luciferase, constitutive promoters

## Abstract

Selecting suitable promoters to drive gene overexpression can provide significant insight into the development of engineered bacteria. In this study, we analyzed the transcriptome data of *Burkholderia pyrrocinia* JK-SH007 and identified 54 highly expressed genes. The promoter sequences were located using genome-wide data and scored using the prokaryotic promoter prediction software BPROM to further screen out 18 promoter sequences. We also developed a promoter trap system based on two reporter proteins adapted for promoter optimization in *B. pyrrocinia* JK-SH007: firefly luciferase encoded by the luciferase gene set (*Luc*) and trimethoprim (*TP*)-resistant dihydrofolate reductase (*TP*^r^). Ultimately, eight constitutive promoters were successfully inserted into the probe vector and transformed into *B. pyrrocinia* JK-SH007. The transformants were successfully grown on Tp antibiotic plates, and firefly luciferase expression was determined by measuring the relative light unit (RLU). Five of the promoters (P4, P9, P10, P14, and P19) showed 1.01–2.51-fold higher activity than the control promoter λ phage transcriptional promoter (PRPL). The promoter activity was further validated via qPCR analysis, indicating that promoters P14 and P19 showed stable high transcription levels at all time points. Then, GFP and RFP proteins were overexpressed in JK-SH007. In addition, promoters P14 and P19 were successfully used to drive gene expression in *Burkholderia multivorans* WS-FJ9 and *Escherichia coli* S17-1. The two constitutive promoters can be used not only in *B. pyrrocinia* JK-SH007 itself to gene overexpression but also to expand the scope of application.

## 1. Introduction

For mammalian humans and their habitat, the *Burkholderia cepacia* complex (Bcc) is a bacterial group that contains both human-unfriendly and human-friendly duality members. Some species can colonize the lungs of patients with cystic fibrosis (CF) for long periods of time, causing chronic opportunistic infections that reduce lung function [1] and make gene therapy more difficult [2]. On the other hand, the natural products of Bcc are an emerging source of various compounds for use in agriculture [3,4,5,6,7] and medicine [5,6,8], such as glycosides, lipopeptides [9], and antibiotics [5,10]. Several strains are known to have biocontrol agents against phytopathogenic fungi, contribute to better water management, and improve nitrogen fixation and plant growth [3]. For instance, *Burkholderia ambifaria* and *Burkholderia caribensis* are presumably diazotrophic strains that promote the growth of the grain crop amaranth [7]. EnacyloxinIIa from *Burkholderia* species was shown to have the most potent antibacterial activity against both Gram-positive and Gram-negative organisms [5]. Currently, tools such as metabolic engineering and synthetic biology are used to discover and produce natural products [4,11,12]. The discovery and exploitation of cloned, native recombinase genes enabled the activation of previously silent BGCs in *Burkholderiales* strain DSM7029, resulting in the isolation of glidopeptin [11]. Examples include heterologous expression of BGCs encoding the lasso peptide capistruin and the polyketide−nonribosomal peptide glidobactin in *E. coli* [12]. *Burkholderia pyrrocinia* JK-SH007, belonging to Bcc, is a strain that excels in promoting plant growth, inhibiting fungal growth [13,14,15], and degrading natural and synthetic contaminants such as the production of siderophores to degrade Fe(III) to usable Fe(II) [14]. Accordingly, engineered bacteria with enhanced antagonistic fungal resistance [16] and insect resistance [17] have been obtained via genetic modification. These two engineered *B. pyrrocinia* JK-SH007-derived strains that we constructed utilize an inducible promoter, namely, the λ phage transcriptional promoter (PRPL), to express the chitinase gene [16] and the *cry218* gene, encoding the insecticidal crystal protein Cry1Ac [17]. PRPL is a temperature-inducible promoter, and in *B. pyrrocinia* JK-SH007, only this promoter has been successfully used, but the expression of some exogenous genes could not be successfully initiated, so the promoter was never explored further in this strain.

The transcriptional process controlling gene expression occurs at the promoter [18,19], and protein production is closely related to promoter strength [20]; therefore, exploration and characterization of novel promoters are essential for applications in bioengineering [21,22,23]. Constitutive promoters have the advantage of being constantly active, not spatiotemporally specific, and not induced by external factors to activate gene expression. Constitutive promoters of varying strengths, for example, can be utilized to fine-tune gene expression levels; thus, route optimization to obtain greater yields of desirable compounds is facilitated by these promoters [24,25,26,27]. Meanwhile, some cryptic gene clusters can be activated via strong constitutive promoters, as a result of which new natural features can be discovered [28,29,30,31]. Therefore, we prioritized the identification of additional constitutive promoters in *B. pyrrocinia*.

Greater mRNA expression in the transcriptome is indicative of a stronger promoter [32]. The more stable the transcript levels of housekeeping genes are, the higher the likelihood they are under the control of constitutive promoters [24,33]. Using RNA-seq-based methods, researchers have obtained endogenous constitutive promoters with different transcriptional strengths to improve protein manufacturing efficiency and save time and money [34,35]. Liu’s group, for instance, investigated *Bacillus licheniformis* atcc14580 transcriptome data to uncover a novel efficient promoter (PBL9) with 23% greater expression than p43inb [34]. Geng’s laboratory [35] identified a root-specific promoter and created a basic digital expression profile for peanuts.

Common probe vectors can be used to screen for promoters. The expression of reporter genes is monitored by cloning uncharacterized DNA fragments into cloning sites at the 5’ end of the promoter-less region and evaluating the promoter strength in the DNA fragments [36,37]. Many genes encode reporter proteins that have been successfully used for quantitative, nondestructive, in vivo transcriptional monitoring [38,39]. A reporter protein can be an enzyme, and the greatest advantage of enzymes is their relatively high sensitivity, as even at low levels, enzymes can catalyze the hydrolysis of enough substrate molecules over time to produce a detectable signal [40]. For example, β-galactosidase hydrolyzes externally provided substrates and produced detectable products [41,42,43,44]. In the presence of ATP, Mg^2+^, and O_2_, firefly luciferase catalyzes the oxidation of luciferin to create the oxidized luciferin oxyluciferin. Bioluminescence is emitted during the oxidation of luciferin, which is then measured by chemiluminescence or the liquid flash assay [45,46]. In addition, the use of genes encoding antibiotic resistance proteins as reporter genes reduce the workload by excluding transformants without initiation function at the resistance plate screening stage.

With the increasing interest in Bcc in agriculture and natural product research, the gradual decrease in the cost of bacterial genomic and transcriptomic sequencing has allowed the identification of dependable constitutive promoters. The promoter sequences in this work were discovered via a thorough investigation of *B. pyrrocinia* JK-SH007 transcriptome sequencing data under various settings. Before the screening, we constructed a promoter probe vector using the pBBR1 plasmid [47] as a background vector with promoter-less LUC and *TP*^r^ as reporter proteins. For experimental assessment, these promoters were identified in JK-SH007, *B. multivorans* WS-FJ9 [48], and *E. coli* S17-1 using trimethoprim (Tp) antibiotic plates and firefly luciferase (LUC) dual reporters as well as real-time quantitative polymerase chain reaction (RT–qPCR). Finally, we have harvested two constitutive promoters that can not only overexpress the fluorescent proteins GFP and RFP in JK-SH007 but also apply to two other bacteria, and we hope that these new constitutive promoters can be used for genetic modification of microorganisms and genetic background studies.

## 2. Results

### 2.1. Screening of Strong Constitutive Promoters in B. pyrrocinia via RNA-Seq

The flow of the experimental method is shown in Figure 1. Constitutive promoters produce relatively steady gene expression patterns, which may be revealed by choosing genes with extremely consistent expression profiles. As a result, bioinformatics approaches were used to examine three transcriptome datasets of *B. pyrrocinia* JK-SH007 from three target media (NA, MSA, and MSA-ZN) and a specific time point (20 h) of growth. The transcriptomes in the three datasets were annotated via the genome of JK-SH007 (to be uploaded), which revealed 6265, 6266, and 6400 genes in the NA, MSA, and MSA-ZN datasets, respectively.

Under a fixed experimental condition, a larger fragments per kilobase of exon model per million mapped fragments (FPKM) value, corresponding to a higher transcript level, indicates a stronger constitutive promoter [49]; this analysis revealed that the expression levels of most genes, accounting for approximately 98% of the total gene abundance, in *B. pyrrocinia* JK-SH007 were low (Figure 1g). Only the top 2.0% high FPKM values (∼126 genes) in each condition were preferred, and 54 genes were highly expressed under all conditions (Figure 1h), indicating that the promoters of these genes may be constitutive.

Based on the *B. pyrrocinia* JK-SH007 genome map, the approximate locations of 54 gene promoters could be determined (Figure 1h). We used the prokaryotic promoter prediction software BPROM to score the first 40 promoter sequences and finally obtained 18 fragments matching the promoter signature (Table 1). Meanwhile, eight sequences were successfully cloned and transferred into JK-SH007 strains, and the sequences are shown in Appendix A.

### 2.2. Evaluation of the Constitutive Promoters in B. pyrrocinia JK-SH007

To further determine whether they are constitutive promoters and promoter strengths, JK-SH007 strains containing these eight sequences were chosen for experimental testing. Reporter genes (*Luc* and *TP*^r^) without promoters served as negative controls (Appendix A), whereas those regulated by the inducible promoter PRPL served as positive controls. Luciferase is produced from the *Luc* reporter gene [50], which has the potential of drastically minimizing interference from autofluorescence; for example, strain CEP559 of the *B. cepacia* complex associated with pMLS7-eGFP exhibited a high level of autofluorescence that interfered with the detection of eGFP-specific fluorescence [51].

Depending on the growth curve of *B. pyrrocinia* JK-SH007 (Appendix A), samples were harvested at the exponential phase (20 h), in which the eight promoter activities were evaluated by determining the relative light units (RLUs) expression in JK-SH007. The activity of the eight promoters varied from 61 to 251% that of PRPL, with five promoters (P4, P9, P10, P14, and P19) having 1.01–2.51-fold higher activity than PRPL (Figure 2a).

In addition, real-time qPCR analysis further assessed the activity of these promoters in *B. pyrrocinia* JK-SH007, which were collected throughout the exponential (20 h), transitional (30 h), and stationary (34 h) phases (Figure 2b). Compared with the results of the firefly luciferase activity assay, at the transcriptional level, five promoters (P4, P9, P10, P14, and P19) showed high transcription at least at one time point; however, the time point for the highest strength of each promoter was different. Exclusively, the promoters P14 and P19 were stably transcribed at all time points, whereas promoters P4, P10, and P9 were less active at all points, which was not consistent with the results of the firefly luciferase assay. Therefore, we concluded that P14 and P19 were the strongest and most stable constitutive promoters in strain *B. pyrrocinia* JK-SH007 under the selected culture conditions.

### 2.3. Assessment of Constitutive Promoters in Different Bacterial Species

We further investigated the strength of these two constitutive promoters (P14 and P19) in *B. multivorans* WS-FJ9 and *E. coli* S17-1 to determine whether they were relevant in additional bacterial species. The experimental designs in these strains were the same as those in *B. pyrrocinia* JK-SH007.

The samples of *E. coli* S17-1 were harvested at the exponential phase (6 h) according to the growth curve of *E. coli* S17-1 (Appendix A), and the firefly luciferase activity was measured. The activities of P14 and P19 were 0.88–2.03 times higher than PRPL (Appendix A). Furthermore, *E. coli* S17-1 was collected in the exponential (6 h), transitional (24 h), and stationary (27 h) phases for real-time qPCR analysis. As shown in Appendix A, at the transcriptional level, these two promoters produced stable and high levels of *Luc* expression at different growth stages in *E. coli* S17-1, which was consistent with the results of the firefly luciferase activity assay.

In addition to *E. coli*, we also tested *B. multivorans* WS-FJ9, a member of the Bcc family. Samples were collected at the exponential phase (6 h) according to the growth curve of WS-FJ9 (Appendix A). The firefly luciferase activity assay showed that the promoters P14 and P19 were 1.45–2.13-fold more active than PRPL (Appendix A). Real-time qPCR analysis was then carried out in the exponential (10 h), transitional (24 h), and stationary (48 h) phases, and the results showed that both promoters produced stable expression at the transcriptional level in *B. multivorans* WS-FJ9 at all three growth stages (Appendix A).

Compared to the inducible strong promoter PRPL, P14 and P19 showed higher activity and could be successfully applied to *B. multivorans* WS-FJ9 and *E. coli* S17-1, suggesting that these promoters may also be used in other bacteria.

### 2.4. Fluorescence Detection in the Engineered B. pyrrocinia JK-SH007 Constructed by Promoters

This study aimed to investigate whether the green fluorescent protein (GFP) and the red fluorescent protein (RFP) are expressed in JK-SH007 transformants under the control of different promoters. Observations were made in bright field and fluorescence mode under a 100× oil immersion lens. Transformants without a promoter and containing PRPL promoter were used as negative control and positive control samples, respectively. We can see the cells under the bright field in Figure 3A–H. Under the dark-field GFP fluorescence filter, transformants containing PRPL, P14, and P19 promoters emitted green fluorescence, as shown in Figure 3F–H. Under the dark-field RFP fluorescence filter, transformants containing PRPL, P14, and P19 promoters emitted red fluorescence (Figure 3N–P), while no fluorescence was found for transformants without promoters, as shown in Figure 3E,M.

### 2.5. Western Blot Analysis in the Engineered B. pyrrocinia JK-SH007 Constructed by Promoters

To further validate the role of promoter in JK-SH007, Western blot analysis was performed. Before spiking, it was ensured that the total protein concentration of each sample was 20 μg and loaded onto SDS-PAGE gels. After electrophoresis and membrane transfer, specific antibodies were used to identify the relative amounts of GFP and RFP. Additionally, the amount of GFP and RFP in JK-SH007 transformants was assessed in a semi-quantitative manner. Western Blot analysis showed that a GFP band of 26 kDa (Figure 4A) and an RFP band of 25 kDa (Figure 4C) were detected using specific antibodies, respectively. Under the same conditions, both GFP and RFP bands were promoter 19, 14, and PRPL in order from dark to light (Figure 4A,C). The GFP and RFP concentrations were in the order of promoter 19, 14, and PRPL from high to low (Figure 4B,D). As shown in Figure 4B, the content of GFP in promoter 19-containing transformants was about 1.84 times higher than that in promoter PRPL-containing transformants; the content of GFP in promoter 14-containing transformants was about 1.36 times higher than that in promoter PRPL-containing transformants; this indicates that promoters 19 and 14 increased the total amount of GFP in JK-SH007 transformants. As shown in Figure 4D, the content of RFP in promoter 19-containing transformants was about 2.54 times higher than that in promoter PRPL-containing transformants; the content of RFP in promoter 14-containing transformants was about 1.43 times higher than that in promoter PRPL-containing transformants; this indicated that promoters 19 and 14 increased the total amount of RFP in JK-SH007 transformants.

The green and red fluorescence observed verifies that both promoters P14 and P19 function in JK-SH007. Western Blot analysis and Bradford protein concentration assay revealed the promoter’s function in JK-SH007 from strong to weak in the order of P19, P14, and the control PRPL.

## 3. Discussion

In a previous laboratory, we constructed these two engineered *B. pyrrocinia* JK-SH007-derived strains using an inducible promoter, the λ phage transcriptional promoter (PRPL), to express the chitinase gene [16] and the *cry218* gene [17]. PRPL is a temperature-inducible promoter that cannot be *B. pyrrocinia* JK-SH007 for stable use. The constitutive promoter S7, derived from the BCC family, provided 5-21-fold GFP expression in *B. cepacia* strains compared to *E. coli*. In addition, the arabinose-induced BAD promoter derived from *E. coli* provided 131-fold higher GFP expression in *B. vietnamiensis* compared to *E. coli* [51]. Promoters derived from the BCC family have been continuously reported, such as the promoter that screens the *Xy1R* gene associated with growth rate to initiate the expression of other proteins [52].

Constitutive promoters are distinguished by the fact that they are unaffected by changes in the life cycle and culture conditions, among other factors, and sustain consistent expression. We conducted this investigation because of a shortage of accessible constitutive promoters for creating engineered strains of B. pyrrocinia JK-SH007. We evaluated the transcriptome data of B. pyrrocinia JK-SH007 and chose the first 2% of genes with high expression levels. Afterward, the promoter sequences of eight genes were determined via promoter prediction using an online tool, and a double reporter gene probe vector for promoter screening was constructed. Finally, a stable promoter was obtained via antibiotic plate selection of positive transformants, luciferase activity detection, and qPCR verification. Based on the RNA-seq method, researchers have obtained native constitutive promoters with different transcriptional strengths. Previously, Wang’s laboratory reported the screening of native promoters from Burkholderia strain DSM 7029, and three promoters were used for heterologous gene expression [49]. Therefore, the strategy used for identifying constitutive promoters in this work is reliable.

Because these transcriptome data were obtained from three target culture conditions and one definite time point, which are insufficient for simulating the mapping of gene transcripts under different conditions, the false-positive rate in selecting promoters was relatively high [53]. Among the eight constitutive promoters tested, there were only two promoters showed higher transcriptional strength than PRPL at all three time points, while the other three promoters (4, 9, and 10) were weaker than the control at one time point at least. The reason for this phenomenon was that we mainly focused on constitutive promoters selected based on genes that were stably expressed at high levels during the growth cycle, while promoters possessing high strength over a short duration were discarded. We speculate that the weaker activity of promoters 15, 17, and 18 was because they represent intergenic regions in the operon context but not constitutive promoter sequences. Several adjacent genes may belong to the same operon region at the same time, and the high expression of the selected genes may be driven by the promoter in front of the whole operon [53].

The protein expression level of a gene correlates with the intensity of the ribosome binding site (RBS). The amount of luciferase activity we observed indicated the role of both promoter and RBS, while the expression profile obtained by qPCR analysis reflected only the role of the corresponding promoter. Although sequences with RBS characteristics were added to the promoter sequences, we did not determine the RBS of each promoter in our experiments, so the qPCR results were not consistent with the fluorescence detection results [49].

Some *B. cepacia* strains exhibit GFP autofluorescence [51] and background interference if GFP is used as a reporter gene. The firefly luciferase reporter gene was used to detect firefly luciferase activity by using firefly luciferin as a substrate. With the help of Mg^2+^ and O_2_, luciferase catalyzes the oxidation of luciferin to oxyluciferin. During the oxidation of luciferin, bioluminescence is emitted, which is then measured chemiluminescence assay. On the other hand, we added the uncommon antibiotic resistance gene *TP*^r^ downstream of the luciferin gene, which greatly reduced false positives. Therefore, screening promoters with dual reporter gene vectors can reduce background interference and improve the screening rate of positive transformants.

We studied whether the foreign genes can be expressed and the amount of expression in engineered bacteria are affected by many factors, such as exogenous genes, expression vectors, host cells, culture conditions, and inducers. Using a luciferase kit assay, we determined the expression level of the exogenous gene *Luc* with eight constitutive promoters at the protein level; therefore, we could replace the *Luc* gene with an exogenous source and select a suitable promoter for exogenous gene expression. The construction of engineered *B. pyrrocinia* JK-SH007-derived strains provides a reference for further improvement of the expression system of exogenous genes. Moreover, we verified the wide applicability of promoters in our study, and the specific constitutive promoters we screened can be applied in *B. multivorans* WS-FJ9 and *E. coli* S17-1. Therefore, the constitutive promoter screening strategy used in this study can provide a reference for the application of a large range of microbial expression vector constructs.

## 4. Materials and Methods

### 4.1. Strains and Reagents

Table 2 lists the wild-type and mutant bacterial strains utilized in this investigation. *E. coli* DH5α was grown in Luria–Bertani (LB) broth (LB; 10 g/L tryptone, 5 g/L yeast extract, 10 g/L NaCl) for molecular cloning and plasmid propagation, with gentamicin (50 μg/mL) added as an antibiotic when needed. *B. pyrrocinia* JK-SH007 and its transformants were cultured in LB medium or agar plates containing suitable antibiotics (50 μg/mL gentamicin and 100 μg/mL trimethoprim (Tp)) at 30 °C. *B. multivorans* WS-FJ9 and its mutants were cultured in LB medium or agar plates containing suitable antibiotics at 30 °C. *E. coli* S17-1 and its mutants were cultured in LB medium or agar plates containing suitable antibiotics at 37 °C. The induction temperature of all mutants containing the PRPL promoter was 42 °C. All plates were prepared by adding 18 g/L agar to the medium. Sangon Biotech in China generated all of the oligonucleotides. New England Biolabs provided restriction enzymes and DNA markers. Sangon Biotech in China supplied the antibiotics and culture medium components (Sangon Biotech, Shanghai, China). Seamless cloning employing the pEASY-Uni Seamless cloning and Assembly Kit (TransGen, Beijing, China) was referred to as the general recombination technique. Firefly Luciferase Reporter Gene Assay Kit (Yeasen, Shanghai, China) was used to help luciferase emit bioluminomescence, containing cell lysis mixture and firefly luciferase assay reagent. Thermo Scientific Multiskan SkyHigh full-wavelength enzyme lab provided bioluminescence assays. Primer design was performed using DNAMAN 6, 0, 3, 99 software. The design principles are as follows: first determine the sequence to be cloned; design forward primers from the head; design reverses primers from the tail; primer length between 15–30 bp, and primers containing homologous arms between 30–45 bp; T_m_ values are generally between 55 °C and 70 °C; the smaller the difference between the T_m_ values of forward and reverse primers by increasing or decreasing the sequence length, the better. The flow of the experimental method is shown in Figure 1.

### 4.2. RNA-Seq Analysis

Transcriptome sequencing was carried out by the Beijing Genomics Institute (BGI) in China. Cells were cultivated at 30 °C for 20 h while being shaken constantly (200 rpm), and samples for RNA isolation were collected in three different media: NA (peptone, 10.0 g; beef powder, 3.0 g; sodium chloride, 5.0g; pH 7.3 ± 0.1), MSA (20 g/L sucrose, 2 g/L asparagine, 1 g/L K_2_HPO_4_, and 0.5 g/L MgSO_4_·7H_2_O), and MSA-ZN (Zn (NO_3_)_2_-6H_2_O was added to NA medium to make the concentration of Zn^2+^ in the medium of 5 mmol/L). RNA was extracted from three biological replicates using Trizol reagent (Ambion, Life Technologies ™, Foster City, CA, USA). Specific procedures were performed according to our study (PRJNA693100) [55]. After sample extraction, the extracted total RNA was first removed from ribosomal RNA (rRNA) and then fragmented. Subsequently, double-stranded cDNA is synthesized. The double-stranded cDNA is then end-repaired, A-tailed, and ligated to sequencing junctions. The ligated products are purified and amplified to obtain the final cDNA library. Finally, the constructed sequencing libraries are sequenced using the HiSeq sequencing platform. The sequenced data are called raw reads or raw data. The information is analyzed as follows: first, the reads with low quality, contaminated junctions, and high content of unknown bases N are removed from the raw data using the NGS QC Toolkit (2.3.3) to obtain clean reads. The clean reads were then aligned to the reference genome sequence using HISAT, and each sample was reconstructed using Rockhopper to obtain new transcripts. The well-known mRNA, novel mRNA, was used as the reference gene set, and the reads were compared to them using Bowtie 2, 3, 5, 1 software, and then the expression levels of the genes were calculated using RSEM v1. 3. 0 software to calculate the expression levels of the genes. To perform the analysis of expression levels, sample correlation was conducted using the cor function in the R 4. 0. 2 software. For linear correlation of the two data sets, Pearson was chosen for the analysis. The *B. pyrrocinia* JK-SH007 genome (PRJNA819416) was used as a reference for transcript identification by Bowtie 2 (Version 2.2.9). Based on the expression results of well-known mRNA and novel mRNA, we analyzed the expression of each sample. log_10_FPKM is represented by FPKM; the higher this value, the higher the gene expression [49]. The promoters can also be scored by ‘prediction of bacterial promoters’ (BPROM) [56,57,58]. The BPROM online software evaluates and scores unknown sequences precisely on the basis of general characteristics of prokaryotic promoters, such as -35 region, -10 region, -35 region, and -10 region with 17 ± 1 bp spacer sequence, etc., to predict the presence of promoters.

### 4.3. Adapting a Promoter-Trapping Vector for Use in B. pyrrocinia

First, we constructed the control plasmid pBBR1-GM^r^-PRPL-Luc-TP^r^ (Appendix A). The 4737-bp pBBR1-GM^r^-PRPL larger fragment was amplified via PCR using primer pairs PT1-F/PT1-R (homology extension for seamless cloning is italicized, Appendix A) and pBBR1-GM^r^ [59] as template. The 1064-bp PRPL smaller fragment was amplified via PCR using primer pairs PRPL-F/PRPL-R (homology extension for seamless cloning is italicized) and pHKT2 [54] as template. The 2033-bp Luc-TP^r^ smaller fragment was obtained via artificial synthesis and amplified via PCR using primer pairs Luc-TP^r^-F/Luc-TP^r^-R (homology extension for seamless cloning is italicized) and sequence (GenBank: MK484106.1) [50] and pHKT2 as templates. The PCR products were separated into 1% agarose gels and extracted using the DNA gel extraction kit (Axygen, Hangzhou, China). After gel-purification, three products were ligated using seamless cloning [60] to yield pBBR1-GM^r^-PRPL-Luc-TP^r^. The 3128-bp PRPL-Luc-TP^r^ region was amplified via PCR using primer pairs PTC1-F/PTC1-R and pBBR1-GMr-PRPL-Luc-TP^r^ as template to verify the correctness of the vector construction.

The promoter regions of 8 highly expressed genes were cloned into the pBBR1-GM^r^-PRPL-Luc-TP^r^ plasmid, which employed the common template of *B. pyrrocinia* JK-SH007 genomic DNA. The primers were designed based on the candidate promoter sequences of *B. pyrrocinia* JK-SH007 predicted via BPROM online software (Table 1), which were used for promoter amplification via polymerase chain reaction (PCR) following the protocol recommended by the manufacturer. The amplification system was 20 μL: 2×EasyTaq^®^ PCR SuperMix at 10 μL and 1 μL of 10 μM primers for each one, 1 μL of the template, and 7 μL of ddH_2_O to fill the volume up to 20 μL. The PCR reaction conditions were denaturing at 94 °C for 5 min and again at 94 °C for 30 s, then annealing at 56 °C for 30 s, extending at 72 °C for 45 s for 35 cycles in total, and then extending at 72 °C for 10 min.

pBBR1-GMr-P4-Luc-TP^r^. The 6836-bp pBBR1-GMr-Luc-TP^r^ larger fragment was amplified via PCR using primer pairs PT2-F/PT2-R and pBBR1-GMr-PRPL-Luc-TP^r^ as templates. The 683-bp P4 promoter fragment was amplified via PCR using primer pairs P4-F/P4-R (homology extension for seamless cloning is italicized) and *B. pyrrocinia* JK-SH007 genome as template. After gel purification, two products were ligated using seamless cloning to yield pBBR1-GMr-P4-Luc-TP^r^. The 972-bp P4 promoter region was amplified via PCR using primer pairs PTC2-F/PTC2-R and pBBR1-GMr-P4-Luc-TP^r^ as template to verify the correctness of the vector construction.

pBBR1-GMr-P9-Luc-TP^r^. The 6836-bp pBBR1-GMr-Luc-TP^r^ larger fragment was amplified via PCR using primer pairs PT2-F/PT2-R and pBBR1-GMr-PRPL-Luc-TP^r^ as template. The 696-bp P9 promoter fragment was amplified via PCR using primer pairs P9-F/P9-R (homology extension for seamless cloning is italicized) and *B. pyrrocinia* JK-SH007 genome as template. After gel purification, two products were ligated using seamless cloning to yield pBBR1-GMr-P9-Luc-TP^r^. The 985-bp P9 promoter region was amplified via PCR using primer pairs PTC2-F/PTC2-R and pBBR1-GMr-P9-Luc-TP^r^ as template to verify the correctness of the vector construction.

pBBR1-GMr-P10-Luc-TP^r^. The 6836-bp pBBR1-GMr-Luc-TP^r^ larger fragment was amplified via PCR using primer pairs PT2-F/PT2-R Rand pBBR1-GMr-PRPL-Luc-TP^r^ as template. The 253-bp P10 promoter fragment was amplified via PCR using primer pairs P10-F/P10-R (homology extension for seamless cloning is italicized) and *B. pyrrocinia* JK-SH007 genome as template. After gel purification, two products were ligated using seamless cloning to yield pBBR1-GMr-P10-Luc-TP^r^. The 542-bp P10 promoter region was amplified via PCR using primer pairs PTC2-F/PTC2-R and pBBR1-GMr-P10-Luc-TP^r^ as template to verify the correctness of the vector construction.

pBBR1-GMr-P14-Luc-TP^r^. The 6836-bp pBBR1-GMr-Luc-TP^r^ larger fragment was amplified via PCR using primer pairs PT2-F/PT2-R and pBBR1-GMr-PRPL-Luc-TP^r^ as template. The 554-bp P14 promoter fragment was amplified via PCR using primer pairs P14-F/P14-R (homology extension for seamless cloning is italicized) and *B. pyrrocinia* JK-SH007 genome as template. After gel purification, two products were ligated using seamless cloning to yield pBBR1-GMr-P14-Luc-TP^r^. The 843-bp P14 promoter region was amplified via PCR using primer pairs PTC2-F/PTC2-R and pBBR1-GMr-P14-Luc-TP^r^ as template to verify the correctness of the vector construction.

pBBR1-GMr-P15-Luc-TP^r^. The 6836-bp pBBR1-GMr-Luc-TP^r^ larger fragment was amplified via PCR using primer pairs PT2-F/PT2-R and pBBR1-GMr-PRPL-Luc-TP^r^ as template. The 260-bp P15 promoter fragment was amplified via PCR using primer pairs P15-F/P15-R (homology extension for seamless cloning is italicized) and *B. pyrrocinia* JK-SH007 genome as template. After gel purification, two products were ligated using seamless cloning to yield pBBR1-GMr-P15-Luc-TP^r^. The 549-bp P15 promoter region was amplified via PCR using primer pairs PTC2-F/PTC2-R and pBBR1-GMr-P15-Luc-TP^r^ as template to verify the correctness of the vector construction.

pBBR1-GMr-P17-Luc-TP^r^. The 6836-bp pBBR1-GMr-Luc-TP^r^ larger fragment was amplified via PCR using primer pairs PT2-F/PT2-R and pBBR1-GMr-PRPL-Luc-TP^r^ as template. The 476-bp P17 promoter fragment was amplified via PCR using primer pairs P17-F/P17-R (homology extension for seamless cloning is italicized) and *B. pyrrocinia* JK-SH007 genome as template. After gel purification, two products were ligated using seamless cloning to yield pBBR1-GMr-P17-Luc-TP^r^. The 765-bp P17 promoter region was amplified via PCR using primer pairs PTC2-F/PTC2-R and pBBR1-GMr-P17-Luc-TP^r^ as template to verify the correctness of the vector construction.

pBBR1-GMr-P18-Luc-TP^r^. The 6836-bp pBBR1-GMr-Luc-TP^r^ larger fragment was amplified via PCR using primer pairs PT2-F/PT2-R and pBBR1-GMr-PRPL-Luc-TP^r^ as template. The 306-bp P18 promoter fragment was amplified via PCR using primer pairs P18-F/P18-R (homology extension for seamless cloning is italicized) and *B. pyrrocinia* JK-SH007 genome as template. After gel purification, two products were ligated using seamless cloning to yield pBBR1-GMr-P18-Luc-TP^r^. The 595-bp P18 promoter region was amplified via PCR using primer pairs PTC2-F/PTC2-R and pBBR1-GMr-P18-Luc-TP^r^ as template to verify the correctness of the vector construction.

pBBR1-GMr-P19-Luc-TP^r^. The 6836-bp pBBR1-GMr-Luc-TP^r^ larger fragment was amplified via PCR using primer pairs PT2-F/PT2-R and pBBR1-GMr-PRPL-Luc-TP^r^ as template. The 307-bp P19 promoter fragment was amplified via PCR using primer pairs P19-F/P19-R (homology extension for seamless cloning is italicized) and *B. pyrrocinia* JK-SH007 genome as template. The PCR products were separated into 1% agarose gels and extracted using the DNA gel extraction kit (Axygen, China). After gel purification, two products were ligated using seamless cloning to yield pBBR1-GMr-P19-Luc-TP^r^. The 596-bp P19 promoter region was amplified via PCR using primer pairs PTC2-F/PTC2-R and pBBR1-GMr-P19-Luc-TP^r^ as template to verify the correctness of the vector construction.

All ligated plasmids were first transformed into *E. coli* DH5α competent cells to obtain a large number of plasmids. Then, all plasmids were chemically transformed into *B. pyrrocinia* JK-SH007. The selected promoter sequences are listed in Appendix A. Finally, the verified plasmid pBBR1-GMr-promoter-luc-TP^r^ was transformed into *B. multivorans* and *E. coli* S17-1 for promoter characterization.

pBBR1-GMr -PRPL-GFP-TP^r^. The 5991-bp pBBR1-GMr-PRPL-TP^r^ larger fragment was amplified via PCR using primer pairs PT3-F/PT3-R and pBBR1-GMr-PRPL-Luc-TP^r^ as template. The 759-bp *gfp* fragment was amplified via PCR using primer pairs *gfp*-F/*gfp*-R (homology extension for seamless cloning is italicized) and pHKT2 as template. After gel purification, two products were ligated using seamless cloning to yield pBBR1-GMr -PRPL-GFP-TP^r^. The 2192-bp *gfp* region was amplified via PCR using primer pairs PTC1-F/PTC1-R and pBBR1-GMr-PRPL-GFP-TP^r^ as template to verify the correctness of the vector construction.

pBBR1-GMr -PRPL-RFP-TP^r^. The 5991-bp pBBR1-GMr -PRPL-TP^r^ larger fragment was amplified via PCR using primer pairs PT3-F/PT3-R and pBBR1-GMr-PRPL-Luc-TP^r^ as template. The 732-bp *rfp* fragment was amplified via PCR using primer pairs *rfp*-F/*rfp*-R (homology extension for seamless cloning is italicized) and PHKT3 [54] as template. After gel purification, two products were ligated using seamless cloning to yield pBBR1-GMr-PRPL-RFP-TP^r^. The 2161-bp *rfp* region was amplified via PCR using primer pairs PTC1-F/PTC1-R and pBBR1-GMr-PRPL-RFP-TP^r^ as template to verify the correctness of the vector construction.

pBBR1-GMr -P14-GFP-TP^r^. The 5695-bp pBBR1-GMr-P14-TP^r^ larger fragment was amplified via PCR using primer pairs PT3-F/PT3-R and pBBR1-GMr-P14-Luc-TP^r^ as template. The 759-bp *gfp* fragment was amplified via PCR using primer pairs *gfp*-F/*gfp*-R (homology extension for seamless cloning is italicized) and pHKT2 as template. After gel purification, two products were ligated using seamless cloning to yield pBBR1-GMr-P14-GFP-TP^r^. The 1896-bp *gfp* region was amplified via PCR using primer pairs PTC1-F/PTC1-R and pBBR1-GMr-P14-GFP-TP^r^ as template to verify the correctness of the vector construction.

pBBR1-GMr -P14-RFP-TP^r^. The 5695-bp pBBR1-GMr-P14-TP^r^ larger fragment was amplified via PCR using primer pairs PT3-F/PT3-R and pBBR1-GMr-P14-Luc-TP^r^ as template. The 732-bp rfp fragment was amplified via PCR using primer pairs *rfp*-F/*rfp*-R (homology extension for seamless cloning is italicized) and PHKT3 as template. After gel purification, two products were ligated using seamless cloning to yield pBBR1-GMr-P14-RFP-TP^r^. The 1866-bp *rfp* region was amplified via PCR using primer pairs PTC1-F/PTC1-R and pBBR1-GMr-P14-RFP-TP^r^ as template to verify the correctness of the vector construction.

pBBR1-GMr -P19-GFP-TP^r^. The 5448-bp pBBR1-GMr-P19-TP^r^ larger fragment was amplified via PCR using primer pairs PT3-F/PT3-R and pBBR1-GMr-P19-Luc-TP^r^ as template. The 759-bp *gfp* fragment was amplified via PCR using primer pairs *gfp*-F/*gfp*-R (homology extension for seamless cloning is italicized) and pHKT2 as template. After gel purification, two products were ligated using seamless cloning to yield pBBR1-GMr-P19-GFP-TP^r^. The 1649-bp *gfp* region was amplified via PCR using primer pairs PTC1-F/PTC1-R and pBBR1-GMr-P19-GFP-TP^r^ as template to verify the correctness of the vector construction.

pBBR1-GMr -P19-RFP-TP^r^. The 5448-bp pBBR1-GMr-P19-TP^r^ larger fragment was amplified via PCR using primer pairs PT3-F/PT3-R and pBBR1-GMr-P19-Luc-TP^r^ as template. The 732-bp *rfp* fragment was amplified via PCR using primer pairs *rfp*-F/*rfp*-R (homology extension for seamless cloning is italicized) and PHKT3 as template. After gel purification, two products were ligated using seamless cloning to yield pBBR1-GMr-P19-RFP-TP^r^. The 1619-bp *rfp* region was amplified via PCR using primer pairs PTC1-F/PTC1-R and pBBR1-GMr-P19-RFP-TP^r^ as template to verify the correctness of the vector construction.

Six ligated plasmids containing *gfp* or *rfp* genes were first transformed into *E. coli* DH5α competent cells to obtain a large number of plasmids. Then, all plasmids were chemically transformed into *B. pyrrocinia* JK-SH007. 

### 4.4. Promoter Characterization via the Quantitative Firefly Luciferase Assay

In total, the remaining 8 promoters were cloned into pBBR1-GM^r^-promoter-Luc-TP^r^ using recombineering (Table 1 and Appendix A). Whether the *B. pyrrocinia* strain generated harbored these recombinants was first determined by plating on TP plates and then verified via PCR (Appendix A). Transformants that obtained all eight promoters could be grown on TP plates. The recombinants of *B. pyrrocinia* JK-SH007 harboring different pBBR1-GM^r^-promoter-Luc-TP^r^ constructs were inoculated into 20 mL of liquid MSA medium and cultured overnight at 30 °C with constant shaking (200 rpm) in a mini shaker (RADOBIO, Shanghai, China). Then, 200 μL of culture was transferred into 20 mL of fresh liquid MSA and cultivated at 30 °C and 200 rpm for 20 h. After entering the fermentation stage, these recombinants continued to be cultured for 8 h at 30 °C, wherein the mutant containing the PRPL promoter was induced at 42 °C.

Cells were collected by centrifugation (Thermo, Raleigh, NC, USA) at the speed of 12,000 g for 2 min at 4 °C and washed twice with PBS (Vazyme, Nanjing, China). At that time, the cell density was adjusted (OD_600_ = 0.8) using NanoDrop 2000C (ThermoFisher Scientific, Waltham, MA, USA), and 90 µL of the culture was mixed with 10 µL of 1 M K_2_HPO_4_ (pH 7.8) and 20 mM EDTA. The mixture was quickly frozen at −80 °C for 10 min before being moved to room temperature to thaw completely. Then, 300 µL of freshly prepared lysis mixture was added, and the sample was incubated for 15 min at room temperature. Afterward, 100 µL each of cell lysate and assay reagent were mixed, and the relative light units (RLUs) were measured using a luminometric assay (PMT voltage: 560, 10 s lag time, 2 s step time, 5 s integration time). Three technical replicates were performed.

### 4.5. Promoter Characterization via qPCR Analysis

For 20 h, *B. pyrrocinia* JK-SH007 strains harboring different pBBR1-GM^r^-promoter-Luc-TP^r^ constructs were cultivated at 30 °C with continual shaking (200 rpm), and samples for RNA isolation were collected in three different media: NA, MSA, and MSA-ZN. Total RNA was extracted using a Bacterial RNA Extraction Kit (Vazyme, Nangjing, China). Gel electrophoresis was used to assess the integrity of the RNA samples, and the quality of the RNA samples was measured via a Thermo Fisher Scientific NanoDrop 2000 spectrophotometer. cDNA was prepared by a HiScript II QRT SuperMix for qPCR Kit (Vazyme, Nangjing, China), including DNA removal treatments by DNase. All primers are listed in Appendix A. Then, 12.5 μL of qPCR Master Mix (Vazyme, Nangjing, China), 2 μL of template cDNA, 0.5 µL of each primer at a concentration of 20 pmol/µL, and 9.5 µL of RNA-free water were mixed gently in each well of a PCR plate. Applied Biosystems 7500 Real-Time PCR System (Applied Biosystems, Waltham, MA, USA) was used to carry out the reactions. Data analysis was performed via the 2^−ΔΔCT^ method. In JK-SH007, the *pyrG* gene, which encodes CTP synthetase, served as an internal control [61]. The 16S ribosomal RNA gene served as an internal control in *B. multivorans* WS-FJ9 and *E. coli* S17-1. Among the three bacteria, Luc was used as the target gene for qPCR experiments. All experiments were conducted independently in triplicate.

### 4.6. Fluorescence Detection in the Engineered B. pyrrocinia JK-SH007 Constructed by Promoters

The plasmid pBBR1-GMr-P14-Luc-TP^r^ was used as a template to clone the vector fragment pBBR1-GMr-P14-TP^r^ with primer PT3-F/R, and pHKT2 was used as a template to clone the insertion fragment *gfp* with primer *gfp*-F/R. The two purified fragments were homologously recombined using a Homologous Recombination Kit to transform the receptor cells *E. coli* DH5α after extracting the plasmid, and this plasmid was pBBR1-GMr-P14-gfp-TP^r^.

The plasmid pBBR1-GMr-P14-Luc-TP^r^ was used as a template to clone the vector fragment pBBR1-GMr-P14-TP^r^ with primer PT3-F/R, and pHKT3 (Tomlin et al., 2004) was used as a template to clone the insertion fragment *rfp* with primer *rfp*-F/R. The same method was used to obtain the plasmid pBBR1-GMr-P14-rfp-TP^r^.

The plasmid pBBR1-GMr-P19-Luc-TP^r^ was used as a template to clone the vector fragment pBBR1-GMr-P19-TP^r^ with primer PT3-F/R, and pHKT2 was used as a template to clone the insertion fragment *gfp* with primer *gfp*-F/R to obtain the plasmid pBBR1-GMr-P19-gfp-TP^r^ in the same way.

The plasmid pBBR1-GMr-P19-Luc-TP^r^ was used as a template to clone the vector fragment pBBR1-GMr-P19-TP^r^ with primer PT3-F/R, and pHKT3 was used as a template to clone the insertion fragment *rfp* with primer *rfp*-F/R. The plasmid pBBR1-GMr-P19-rfp-TP^r^ was obtained in the same way.

The plasmid pBBR1-GMr-PRPL-Luc-TP^r^ was used as template, the vector fragment pBBR1-GMr-PRPL-TP^r^ was cloned with primer PT3-F/R, the insertion fragment *gfp* was cloned with primer *gfp*-F/R using pHKT2 as template, and the plasmid pBBR1-GMr-PRPL-gfp-TP^r^ was obtained in the same way.

The plasmid pBBR1-GMr-PRPL-Luc-TP^r^ was used as a template to clone the vector fragment pBBR1-GMr-PRPL-TP^r^ with primer PT3-F/R, and pHKT3 was used as a template to clone the insert fragment *rfp* with primer *rfp*-F/R. The plasmid pBBR1-GMr-PRPL-rfp-TP^r^ was obtained in the same way.

Finally, the obtained plasmids were transformed into *B. pyrrocinia* JK-SH007 after verification by sequencing with primer PTC1-F/R.

Cells were visualized via bright field and fluorescent microscopy using a Zeiss Axio Imager M2 Fluorescent microscope (Zeiss, Oberkochen, Germany) under a 100× oil immersion lens. *B. pyrrocinia* JK-SH007 transformants containing the PRPL, P14, and P19 promoters were incubated at 30 °C until OD_600_ reached 0.6, while those containing the PRPL promoter were rapidly raised to 42 °C and incubated for 8 h. Transformants containing the P14 and P19 promoters were incubated at 30 °C for 8 h. All transformants were centrifuged for 2 mL each at 10,000 rpm for 2 min at room temperature, then the supernatant was removed, and the cell sediment was resuspended and dispersed uniformly in 2 mL of a solution containing 50% glycerol and PBS. The cell density of the different samples was adjusted to the same level by measuring OD_600_ and the appropriate dilution. Then, 5 µL of the sample was dropped onto a slide, and the fluorescence signal in JK-SH007 cells was observed and captured using a fluorescence microscope.

### 4.7. Western Blot Analysis in the Engineered B. pyrrocinia JK-SH007 Constructed by Promoters

To further confirm the GFP and RFP levels in the promoter-bearing transformants of JK-SH007, Western analysis was performed to assess the amount of GFP and RFP semi-quantitatively. The 50 mL of bacterial broth after induction in the fluorescence detection assay was centrifuged at 10,000 rpm for 10 min at 4 °C, and the precipitate was resuspended in 5 mL of lysis buffer (2.5 M KCl, 50 mM MgCl_2_, 1 mM EDTA, 5% [*v*/*v*] glycerol, and 50 mM Tris-HCl pH 8.0). These mixtures were then individually sonicated by sonicator (Jingxin, Xinchang, China) on ice for 15 min and boiled in water for 5 min to obtain total protein. Protein concentrations were determined using the Bradford Protein Assay Kit (Beyotime Biotechnology, Shanghai, China). Total proteins were first separated by SDS-PAGE Analysis of 12% gel kit (Beyotime Biotechnology, Shanghai, China) and then transferred to polyvinylidene fluoride (PVDF) (Wissen, Shenzhen, China) membrane using Western Blot kit (Beyotime Biotechnology, Shanghai, China). The blocking solution was 1% bovine serum albumin (BSA) (Thermo Fisher Scientific, Waltham, MA, USA) in 1×PBST (Sangon Biotech, Shanghai, China) configuration. Primary and secondary antibodies were diluted separately with the blocking solution. Blots were blocked with 1% BSA for 1 h at room temperature, incubated with primary antibody of Anti-GFP antibody (1:10,000, Abcam, New York, NY, USA) overnight at 4 °C, and then incubated with the secondary antibody of Goat Anti-Rabbit IgG (Alkaline Phosphatase) (1:1000, Abcam, New York, NY, USA) in secondary antibody incubation solution and incubated for 1h at room temperature on a shaker. The primary antibody for Anti-RFP (1:1000, Affinity, Golden, CO, USA) was incubated overnight at 4 °C, followed by incubation with Goat Anti-Rabbit IgG H&L(HRP) secondary antibody (1:5000, Abcam, New York, NY, USA) for 1 h at room temperature. After the blocking process, the two incubation processes were followed by three separate washes with 1× PBST and the blocking and incubation processes were all stirred left and right on a shaker (Jarell, Wuxi, China). ECL luminescent solution (Super Signal™ West Femto Maximum Sensitivity Substrate) was used to detect the substrate; the signal was observed by X-ray compression and washing of the film.

### 4.8. Statistical Analysis 

The statistical significance of differences was analyzed using Student’s *t*-test in Prism 5 software (GraphPad Software, Inc., San Diego, CA, USA). The data were shown as means ± SD. Differences with *p*-values < 0.05 were considered statistically significant.

## 5. Conclusions

In summary, by using RNA-seq technology, we identified eight candidate promoter regions from *B. pyrrocinia* JK-SH007. Promoter probe vectors with two reporter proteins, firefly luciferase encoded by the luciferase gene set (*Luc*) and Tp-resistant dihydrofolate reductase (*TP*^r^), were used for rapid detection of promoter competence. To demonstrate their utility, 8 of these 16 promoters were successfully used to initiate the expression of dual reporter proteins. They were successfully grown on Tp antibiotic plates and exhibited luciferase expression, with five of the promoters having higher activity than the control. The promoter activity was further verified via qPCR analysis, which showed that the two promoters showed stable high transcript levels at all time points. The P14 and P19 promoters enable the expression of GFP and RFP proteins in *B. pyrrocinia* JK-SH007, prompting us to continue to apply them in the future construction of engineered bacteria of JK-SH007. Besides, P14 and P19 promoters were experimentally evaluated in *B. multivorans* WS-FJ9, implying that these two promoters may be reliable constitutive promoters and can be extended to be applied to other strains in the Bcc. Experimental evaluation in *E. coli* S17-1 suggests that these two are constitutive promoters of broad strength that may be valuable in metabolic engineering and synthetic biology. This study demonstrates the potential application of probe vector systems for enzyme reporter proteins and antibiotic resistance proteins in verifying bacterial promoter activity. Furthermore, the methodical and reasonable promoter technique used in this work may serve as a model for promoter mining in other species.

## Figures and Tables

**Figure 1 ijms-24-09419-f001:**
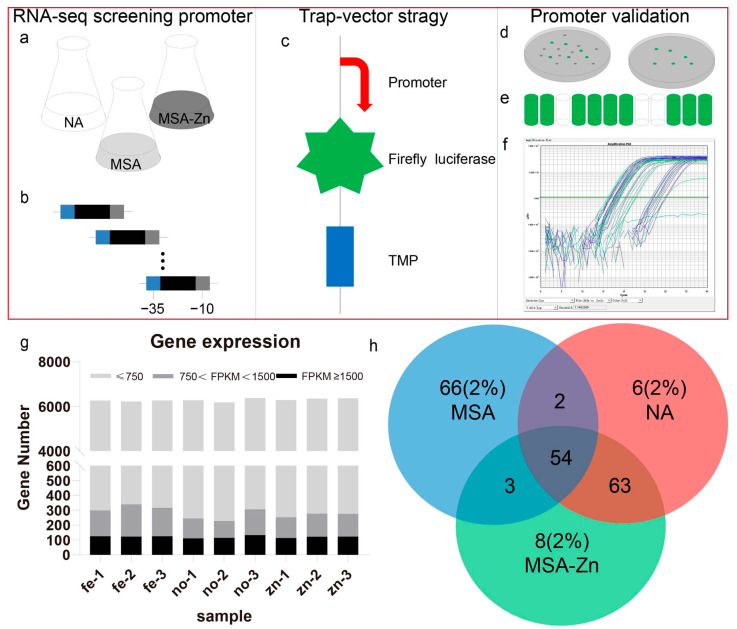
The procedure for screening, evaluating, and verifying new constitutive promoters from *B. pyrrocinia* JK-SH007 is depicted schematically. (**a**) We obtained RNA-seq data from *B. pyrrocinia* JK-SH007 in three media of NA, MSA, and MSA-Zn. (**b**) We chose a promoter that exhibited the desired properties. (**c**) The double reporter gene verification vector was created. The promoter fragments were cloned into a promoter trap system, allowing promoters to be screened. (**d**) We manually screened all clones on a trimethoprim (TP) plate; (**e**) evaluated all green, fluorescent clones using a microplate reader; and (**f**) validated promoter activity at the transcriptional level. We were able to uncover and use the new promoters in bacterial engineering because of this integrated technique. (**g**) Distribution of FPKM values from RNA-seq data. The expressed genes in each sample were categorized based on FPKM values ranging between 750 and 1500. fe-1, fe-2, and fe-3 are three independent replicates of the transcriptome data obtained from *B. pyrrocinia* JK-SH007 grown in MSA medium. no-1, no-2, and no-3 are three independent repeats and are transcriptome data obtained from *B. pyrrocinia* JK-SH007 grown in NA medium. zn-1, zn-2, and zn-3 are three independent repeats, which are the transcriptome data obtained from *B. pyrrocinia* JK-SH007 grown in MSA-Zn medium. Light gray is the number of genes with FPKM values not greater than 750; dark gray is the number of genes with FPKM values between 750–1500; black is the number of genes with FPKM values not less than 1500. (**h**) Venn diagram illustrating the number of highly expressed genes (cutoff 320, 2%) based on FPKM values obtained from RNA-seq under three settings.

**Figure 2 ijms-24-09419-f002:**
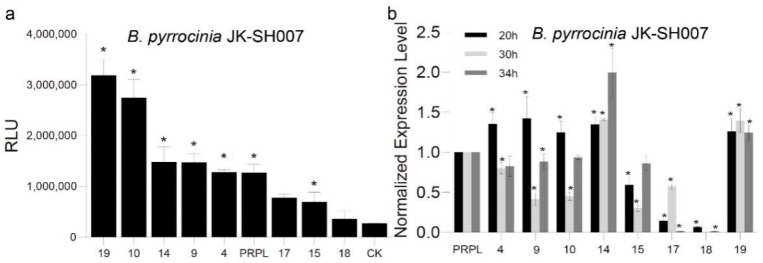
(**a**) Assay of constitutive promoters from *B. pyrrocinia* JK-SH007 using firefly luciferase as reporter protein. CK was a luciferase-specific negative control. The standard deviations from four separate replicates are indicated by the error bars. The data represent means ± SD (*n* = 4). “*” means the amount of firefly luciferase significantly (*p* < 0.05) compared to control promoter PRPL. (**b**) Promoter characterization via qPCR analysis in *B. pyrrocinia* JK-SH007. In three different periods, the transcription of the firefly luciferase gene as target gene under strong promoters in JK-SH007 was measured. The y-axis depicts the expression value of PRPL, which was set to 1. The standard deviations from three separate replicates are indicated by the error bars. The data represent means ± SD (*n* = 4). * *p* < 0.05.

**Figure 3 ijms-24-09419-f003:**
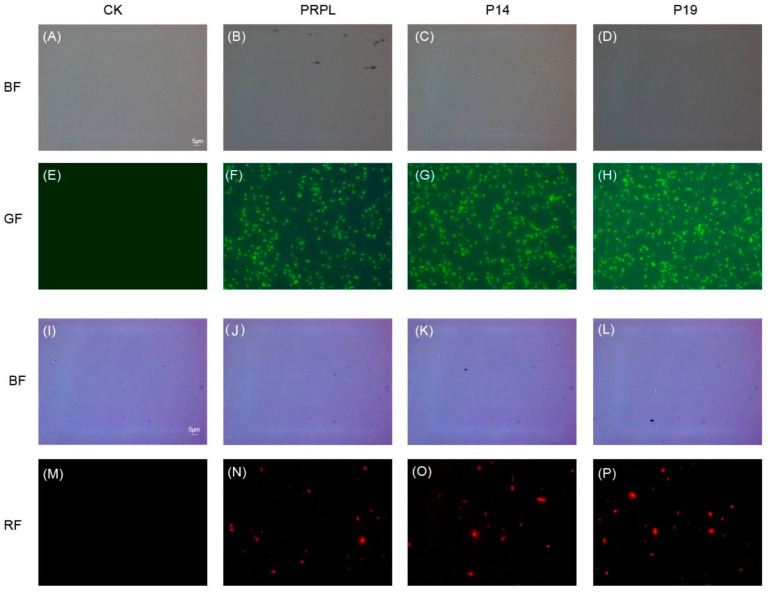
The green and red fluorescence of JK-SH007 bacteria containing different promoters were observed under Zeiss fluorescence microscope. (**A**–**D, I**–**L**) are cells under bright field (BF); (**E**–**H**) are cells under GFP fluorescence filter (GF); (**M**–**P**) are cells under RFP fluorescence filter (RF). (**A**,**E**,**I**,**M**) are cells without promoter; (**B**,**F**,**G**) and (**N**) are cells containing PRPL promoter; (**C**,**G**,**K**) and (**O**) are cells containing P14 promoter; (**D**,**H**,**L**) and (**P**) are cells containing P19 promoter. Scale bar is 5 µm.

**Figure 4 ijms-24-09419-f004:**
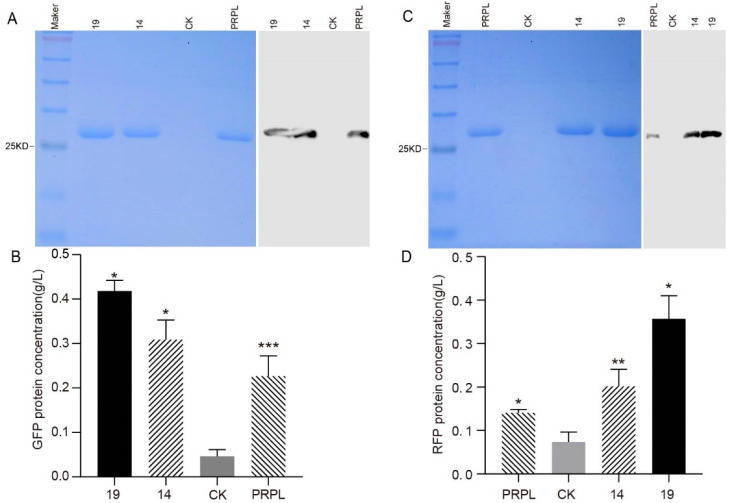
Evaluation of GFP and RFP fluorescent protein expression in JK-SH007 containing different promoters by SDS-PAGE and Western Blot Analysis. CK does not contain a promoter. (**A**) The expression of GFP fluorescent protein containing different promoters was analyzed by SDS-PAGE and Western blotting; (**B**) Semiquantitative analysis of GFP protein expression concentration; (**C**) The expression of RFP fluorescent protein was analyzed by SDS-PAGE and Western blotting; (**D**) Semiquantitative analysis of RFP protein expression concentration. The data represent means ± SD (*n* = 3). Compared to the control CK, “*”, “**”, and “***” means the amount of firefly luciferase significantly (*p* < 0.05), (*p* < 0.01), and (*p* < 0.001), respectively.

**Table 1 ijms-24-09419-t001:** The length of 18 promoter regions and prediction scores via BPROM.

Promoter	Gene ID	Annotation	Length of Promoter (bp)	LDF
P4	JK-SH007-SCAF1GL000497	Outer membrane protein	641	4.23
P6	JK-SH007-SCAF2GL000587	Crp/FNR family transcriptional regulator	193	0.46
P9	JK-SH007-SCAF2GL001878	Putative uncharacterized protein	654	2.50
P10	JK-SH007-SCAF2GL002450	Acyl-homoserine-lactone synthase	211	2.05
P13	JK-SH007-SCAF4GL000338	hypothetical protein	313	4.46
P14	JK-SH007-SCAF1GL002576	phasin family protein	512	4.42
P15	JK-SH007-SCAF1GL000041	dinucleoside polyphosphate hydrolase	218	3.94
P17	JK-SH007-SCAF1GL000053	Putative uncharacterized protein	437	2.13
P18	JK-SH007-SCAF2GL000951	Cysteine sulfinate desulfinase	264	4.23
P19	JK-SH007-SCAF1GL001608	Uncharacterized conserved protein	265	3.70
P20	JK-SH007-SCAF1GL002619	RNA polymerase factor sigma-32	360	2.44
P23	JK-SH007-SCAF1GL002874	Dipeptide-binding ABC transporter, periplasmic substrate-binding component	131	3.45
P24	JK-SH007-SCAF1GL001670	alkyl hydroperoxide reductase/Thiol specific antioxidant/ Mal allergen	198	4.54
P25	JK-SH007-SCAF1GL000981	XRE family transcriptional regulator	489	1.91
P28	JK-SH007-SCAF1GL001828	FKBP-type peptidyl-prolylcis-trans isomerases 2-like protein	113	2.48
P30	JK-SH007-SCAF1GL000139	ribosomal protein L13	209	3.34
P36	JK-SH007-SCAF1GL000818	FtsH peptidase	205	0.43
P39	JK-SH007-SCAF1GL001518	adenylosuccinate synthetase	161	2.41

LDF: Linear discriminant function.

**Table 2 ijms-24-09419-t002:** Strains and plasmids in this work.

Strains	Description	Source
*E. coli* DH5α	Chemically competent cell for cloning	Laboratory stock
*Burkholderia pyrrocinia* JK-SH007	Bcc genomovar IX, an endophytic bacterium isolated from poplar	[13]
*Burkholderia multivorans* WS-FJ9	Bcc genomovar II, a phosphate-solubilizing bacterium	[48]
*E. coli* S17-1	S17-1 λpir chemically competent cell	Laboratory stock
**Plasmids**	**Characteristics**	**Source**
pHKT2	TP^r^, PRPL lambda phage promoters, *gfp* reporter gene	[54]
pBBR1MCS-5	Gm^r^, LacZ alpha peptide	Laboratory stock
pBBR1-GM^r^-PRPL-Luc-TP^r^	TP^r^, Gm^r^, pBBR1 ori, *bla* gene encoding b-lactamase, *rep* gene for plasmid replication, *rrnBT1* transcriptional terminator, PRPL lambda phage promoters, *atpE* translation initiation region, *luc* gene encoding firefly luciferase	This study
pBBR1-GM^r^-P4-Luc-TP^r^	pBBR1-GM^r^-PRPL-Luc-TP^r^ derived, PRPL promoter replaced by the P4 promoter	This study
pBBR1-GM^r^-P6-Luc-TP^r^	pBBR1-GM^r^-PRPL-Luc-TP^r^ derived, PRPL promoter replaced by the P6 promoter	This study
pBBR1-GM^r^-P9-Luc-TP^r^	pBBR1-GM^r^-PRPL-Luc-TP^r^ derived, PRPL promoter replaced by the P9 promoter	This study
pBBR1-GM^r^-P10-Luc-TP^r^	pBBR1-GM^r^-PRPL-Luc-TP^r^ derived, PRPL promoter replaced by the P10 promoter	This study
pBBR1-GM^r^-P13-Luc-TP^r^	pBBR1-GM^r^-PRPL-Luc-TP^r^ derived, PRPL promoter replaced by the P13 promoter	This study
pBBR1-GM^r^-P14-Luc-TP^r^	pBBR1-GM^r^-PRPL-Luc-TP^r^ derived, PRPL promoter replaced by the P14 promoter	This study
pBBR1-GM^r^-P15-Luc-TP^r^	pBBR1-GM^r^-PRPL-Luc-TP^r^ derived, PRPL promoter replaced by the P15 promoter	This study
pBBR1-GM^r^-P17-Luc-TP^r^	pBBR1-GM^r^-PRPL-Luc-TP^r^ derived, PRPL promoter replaced by the P17 promoter	This study
pBBR1-GM^r^-P18-Luc-TP^r^	pBBR1-GM^r^-PRPL-Luc-TP^r^ derived, PRPL promoter replaced by the P18 promoter	This study
pBBR1-GM^r^-P19-Luc-TP^r^	pBBR1-GM^r^-PRPL-Luc-TP^r^ derived, PRPL promoter replaced by the P19 promoter	This study
pBBR1-GM^r^-P20-Luc-TP^r^	pBBR1-GM^r^-PRPL-Luc-TP^r^ derived, PRPL promoter replaced by the P20 promoter	This study

TP^r^, trimethoprim resistance; Gm^r^, gentamicin resistance.

## Data Availability

The transcriptome data used in this current study have been deposited in the GenBank database under record number (PRJNA693100). For genomic datasets, reasonable requests can be made to the corresponding author.

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
