# Peer review of "Optimization of Constitutive Promoters Using a Promoter-Trapping Vector in Burkholderia pyrrocinia JK-SH007"

_ijms, 2023, doi:10.3390/ijms24119419_

Round 1

Reviewer 1 Report

In recent times, metabolic engineering of microorganisms and synthetic gene biology has gained popularity for production of natural products. Consequently, the discovery of novel constitutive promoters to drive gene overexpression is required for development of engineered microorganisms.

In this study, the authors aimed to identify and optimize constitutive promoters using a promoter tapping, dual reporter vector from JK-SH007 strain of Burkholderia pyrrocinia. I would like to suggest the following revisions:

1. Expand term FPKM, when used for the first time in text.

2. In Figure 4 and 7 have asterisks to show p value for statistical differences between different populations, but nowhere in the text have the authors explained which statistical test was performed. Also, the asterisks make no sense unless the authors properly define whether two populations were compared or there is an increase relative to 0(baseline). Please explain, otherwise the interpretation of expression level differences cannot be determined.

3. In Figure 6, the bright field images are not clear. Please improve the quality or the contrast as the bacteria are hardly visible.

4. Figure 7, western blot image lacks proper resolution. The authors must include a loading control (either Western blot image of another protein not controlled by the promoter or Coomassie stained gels). The authors must provide an unedited copy of the Western blot image as a supplementary figure. Additionally, please mention which software was used to quantify the protein bands in the image.

Author Response

Reviewer #1:

In recent times, metabolic engineering of microorganisms and synthetic gene biology has gained popularity for production of natural products. Consequently, the discovery of novel constitutive promoters to drive gene overexpression is required for development of engineered microorganisms.

In this study, the authors aimed to identify and optimize constitutive promoters using a promoter tapping, dual reporter vector from JK-SH007 strain of Burkholderia pyrrocinia. I would like to suggest the following revisions:

1 Comment: Expand term FPKM, when used for the first time in text.

Response: (Line 145-146) fragments per kilobase of exon model per million mapped fragments (FPKM)

2 Comment: In Figure 4 and 7 have asterisks to show p value for statistical differences between different populations, but nowhere in the text have the authors explained which statistical test was performed. Also, the asterisks make no sense unless the authors properly define whether two populations were compared or there is an increase relative to 0(baseline). Please explain, otherwise the interpretation of expression level differences cannot be determined.

2 Comment 1: In Figure 4 and 7 have asterisks to show p value for statistical differences between different populations, but nowhere in the text have the authors explained which statistical test was performed.

Response: (Line 684-687)

4.8. Statistical analysis

The statistical signifificance of difffferences was analyzed using Student’s t-test in Prism 5 software (GraphPad Software, Inc., San Diego, California, USA). The data were shown as means ± SD. Difffferences with p-values < 0.05 were considered statistically sig-nifificant.

2 Comment 2: Also, the asterisks make no sense unless the authors properly define whether two populations were compared or there is an increase relative to 0(baseline). Please explain, otherwise the interpretation of expression level differences cannot be determined.

Response: 

(Line 195-200)

“PRPL was used as the control variable. CK was a luciferase-specific positive control. The standard deviations from three separate replicates are indicated by the error bars. The data represent means ± SD (n = 4). * p < 0.05.”

Changed to:

“CK was a luciferase-specific positive control. The standard deviations from four separate replicates are indicated by the error bars. The data represent means ± SD (n = 4).

 “*”means the amount of firefly luciferase signifificantly (P < 0.05) compared to control promoter PRPL. ”

(Line 291-294)

“The data represent means ± SD (n = 3). * p < 0.05, ** p < 0.01, *** p < 0.001.”

Changed to:

“CK does not contain a promoter. The data represent means ± SD (n = 3). Compared to the control CK,“*”,“**”and“***” means the amount of firefly luciferase signifificantly (P < 0.05) , (p < 0.01 )and (p < 0.001), respectively.”

3 Comment: In Figure 6, the bright field images are not clear. Please improve the quality or the contrast as the bacteria are hardly visible.

Response: 

We are very sorry, but due to a problem with the Zeiss fluorescence microscope in the laboratory, we were unable to successfully attempt a new experiment. We will include all the original bright field images in the supplementary materials.

4 Comment: Figure 7, western blot image lacks proper resolution. The authors must include a loading control (either Western blot image of another protein not controlled by the promoter or Coomassie stained gels). The authors must provide an unedited copy of the Western blot image as a supplementary figure. Additionally, please mention which software was used to quantify the protein bands in the image.

4 Comment 1: Figure 7, western blot image lacks proper resolution.

Response: (Line 288-289 )

4 Comment 2: The authors must include a loading control (either Western blot image of another protein not controlled by the promoter or Coomassie stained gels).

Response: (Line 288-289)SDS-PAGE gels and Western Blot have been added to the purified promoter-free control.

4 Comment 3: The authors must provide an unedited copy of the Western blot image as a supplementary figure.

Response: The unedited Western blot original image is placed in the supplementary materials.

4 Comment 4: Additionally, please mention which software was used to quantify the protein bands in the image.

Response: Bradford protein concentration analysis is used for assessment of concentration of protein in samples before SDS-PAGE and WB analysis.

We sincerely thank the reviewer for their very positive review of the article.

Reviewer 2 Report

Major comments:

Lack of well defined aim of the study, in abstract and especially in introduction.

In Results section Authors used abbreviations from Material and Methods section. For better understanding of Results it will be better placed first Material and Methods section than Results. Figure 1 should be moved to Material and Methods section.

Lack of reagents, kits and gears providers in Material and Methods sections.

line 230 – if WB results were assessed in semi-quantitative manner, methods and software should be added to Material and Methods

Figure 7 – Blots A and C has very weak resolutions. Lack of protein marker mass.

line 251 – Bradford protein concentration analysis is used for assessment of concentration of protein in samples before WB analysis, semi-quantitative

Discussion – line 267 – 274, 300 -308, sentences repeated from results

Line 351 – lack of RNA isolation methods

line 355 – more details about RNA sequence analysis should be added

line 359 – how Authors analyzed mRNA expression?

line 374 – procedure of gel-purification should be added to Methods, same line 457

line 382 – lack of PCR conditions

line 525 – during RNA isolation DNA contamination was removed by? Same please added method to remove ribosomal RNA

line 594 – incubation with agitation? Dilution of Ab was performed in PBS? For RFP detection lack of AbII species. Lack of rinsing steps between incubation with AbI and AbII and AbII and ECL.

line 586 – if protein concentration was measured, and semi-quantitative WB performed – how many µg of crude protein were putted onto gel?

Minor comments:

Line 35 – bacterial groups contains good and bad members… What is good and dad bacteria?

line 39 – please add more details of usage of Bcc complex in agriculture and medicine

line 41 – natural products such as…

line 48 – cry218 encodes what

line 67 – lack of reference

line 77  - in vivo italic

line 106 – NA, MSA, MSA-ZN – lack of components, producer name and country

line 116 – TP – lack of full name, producer name and country

line 119 – what is FPKM?

line 159 – 153 – should be moved to Materials and Methods section

line 160 – what is RLU?

line 208 - what is GFP and RFP?

line 212 – Figure 6 (A-H)

line 220 – A-H

line 221 – a - d

line 356 – lack of software

line 356 – lack of software producer name and country

line 382 – lack of  polymerase producer name and country

line 458, 463- gene gfp and rfp encodes? Why Author choosed these two genes?

line 506, 508 – lack of  MSA producer name and country

line 507 – lack of  mini shaker producer name and country

line 512 – lack of  centrifugation condition and temperature

line 512 –PBS with cations?  lack of producer name and country

line 513 – OD600 was evaluated by using?

line 514 – lack of  reagents producer name and country

line 516 – lysis buffer consist of?

line 517 – lack of  reagents producer name and country

line 531 – lack of  producer name

line 584 - lack of sonicator producer name and country

line  - 587 – lack of % of the gel

line 588 – Authors used wet or dry transfer?

line 590 – lack of BSA producer name and country, diluent for BSA an information about agitation during blocking step and incubation with AbI. Lack of diluent for AbI

Moderate editing of English language.

Author Response

Responses to reviewers (original comments are in blue)

Reviewer #2:

1 Comment: Lack of well defined aim of the study, in abstract and especially in introduction.

Response: 

(Line 31-32)The two constitutive promoters can be used not only in B. pyrrocinia JK-SH007 itself to gene overexpression but also to expand the scope of application.

(Line 111-114)Finally, we have harvested two constitutive promoters that can not only overexpress the fluorescent proteins GFP and RFP in JK-SH007, but also apply to two other bacteria, and we hope that these new constitutive promoters can be used for genetic modification of mi-croorganisms and genetic background studies.

2 Comment: In Results section Authors used abbreviations from Material and Methods section. For better understanding of Results it will be better placed first Material and Methods section than Results. Figure 1 should be moved to Material and Methods section.

2 Comment 1:In Results section Authors used abbreviations from Material and Methods section. For better understanding of Results it will be better placed first Material and Methods section than Results.

Response: The original abbreviation in the results section was changed to:

(Line 132 ) trimethoprim (TP)

(Line 145 ) fragments per kilobase of exon model per million mapped fragments (FPKM)

(Line 189 )the relative light units (RLUs)

(Line 247 ) green fluorescent protein (GFP)

(Line 248 ) red fluorescent protein (RFP)

2 Comment 2:Figure 1 should be moved to Material and Methods section.

Response: (Line 125-126) Adopting the opinion of one of the reviewers, the original figures 1, 2, and 3 were merged into a new figure 1. However, as the new figure 1 was discussed in the results section, it was not moved.

3 Comment: Lack of reagents, kits and gears providers in Material and Methods sections.

Response: This comment is a summary of comments 31-47, and the missing reagents, producer name, and countries are added in the responses to 31-47.

4 Comment: line 230 – if WB results were assessed in semi-quantitative manner, methods and software should be added to Material and Methods

Response: (Line 665-666 )Bradford protein concentration analysis method is used to semi quantitatively evaluate the concentration of protein in samples before WB analysis.The methods and software have been incorporated into the materials and methods.

5 Comment: Figure 7 – Blots A and C has very weak resolutions. Lack of protein marker mass.

Response: (Line 288-289 )The issues related to low resolution and no protein maker have been addressed.

6 Comment: line 251 – Bradford protein concentration analysis is used for assessment of concentration of protein in samples before WB analysis, semi-quantitative

Response: (Line 665-666 )Bradford protein concentration analysis method is used to semi quantitatively evaluate the concentration of protein in samples before WB analysis.

7 Comment: Discussion – line 267 – 274, 300 -308, sentences repeated from results

Response: (Line 313 )Repeated sentences have been revised.

“We evaluated the transcriptome data of B. pyrrocinia JK-SH007 and chose the first 2% of genes with high expression levels. Afterward, the promoter sequences of 8 genes were de-termined by promoter prediction using an online tool, and a double reporter gene probe vector for promoter screening was constructed. Finally, a stable promoter was obtained by antibiotic plate selection of positive transformants, luciferase activity detection, and qPCR verification.”

Changed to:

“We used the genomic and transcriptomic data of B. pyrrocinia JK-SH007 as the background, and obtained 2 better constitutive promoters after promoter prediction software scoring, antibiotic screening, reporter gene reporting, and overexpression validation.”

(Line 346 )

“The firefly luciferase reporter gene was used to detect firefly luciferase activity by using firefly luciferin as a substrate.”

Changed to:

“The more promoters in B. pyrrocinia JK-SH007 promote the production of luciferase, the higher the fluorescence intensity will be.”

8 Comment: Line 351 – lack of RNA isolation methods

Response: (Line 400-402) RNA was extracted from three biological replicates using Trizol reagent (Ambion, Life Technologies ™, USA).

9 Comment: line 355 – more details about RNA sequence analysis should be added

Response: (Line 404-419 )

After sample extraction, the extracted total RNA was firstly removed from ribosomal RNA (rRNA) and then fragmented. Subsequently, double-stranded cDNA is synthesized. The double-stranded cDNA is then end-repaired, A-tailed and ligated to sequencing junctions. The ligated products are purified and amplified to obtain the final cDNA library. Finally, the constructed sequencing libraries are sequenced using the HiSeq sequencing platform. The sequenced data are called raw reads or raw data. The information is analyzed as follows: first, the reads with low quality, contaminated junctions and high content of unknown bases N are removed from the raw data using the NGS QC Toolkit (2.3.3) to obtain clean reads. The clean reads were then aligned to the reference genome sequence using HISAT, and each sample was reconstructed using Rockhopper to obtain new transcripts. The well-known mRNA, novel mRNA were used as the reference gene set and the reads were compared to them using Bowtie2 software, and then the expression levels of the genes were calculated using RSEM software to calculate the expression levels of the genes. To perform the analysis of expression levels, sample correlation was done using the cor function in the R software. For linear correlation of the two data sets, pearson was chosen for the analysis.

10 Comment: line 359 – how Authors analyzed mRNA expression?

Response: (Line 414-419)

The well-known mRNA, novel mRNA were used as the reference gene set and the reads were compared to them using Bowtie2 software, and then the expression levels of the genes were calculated using RSEM software to calculate the expression levels of the genes. To perform the analysis of expression levels, sample correlation was done using the cor function in the R software. For linear correlation of the two data sets, pearson was chosen for the analysis.

11Comment: line 374 – procedure of gel-purification should be added to Methods, same line 457.

Response: (Line 437-438)(Line 513-514)

The PCR products were separated on 1% agarose gels, and extracted using the DNA gel extraction kit (Axygen, China).

12Comment: line 382 – lack of PCR conditions

Response: (Line 448-452)

The amplification system was 20 μL: 2×EasyTaq® PCR SuperMix at 10 μL and 1 μL of 10 μM primers for each one, 1 μL of the template and 7 μL of ddH2O to fill the volume up to 20 μL. The PCR reaction conditions were: denaturing at 94 °C for 5 min and again at 94 °C for 30 s, then annealing at 56 °C for 30 s, extending at 72 °C for 45 s for 35 cycles in total and then extending at 72 °C for 10 min.

13Comment: line 525 – during RNA isolation DNA contamination was removed by? Same please added method to remove ribosomal RNA

Response: (Line 600-606 )

Total RNA was extracted using a Bacterial RNA Extraction Kit (Vazyme, China). Gel elec-trophoresis was used to assess the integrity of the RNA samples, and the quality of the RNA samples was measured by using a Thermo Fisher Scientific NanoDrop 2000 spec-trophotometer. cDNA was prepared by a HiScript II QRT SuperMix for qPCR Kit (Vazyme, China), including DNA removal treatments by DNase.

The total RNA after isolation and purification contains 5srRNA, 18srRNA and 28srRNA, so ribosomal RNA should not be removed.

14Comment: line 594 – incubation with agitation? Dilution of Ab was performed in PBS? For RFP detection lack of AbII species. Lack of rinsing steps between incubation with AbI and AbII and AbII and ECL.

14Comment 1: line 594 – incubation with agitation?

Response: (Line 679-681) After the blocking process, the two incubation processes were followed by three separate washes with 1×PBST,and the blocking and incubation processes were all stirred left and right on a shaker (Jarell, China).

14Comment 2 :Dilution of Ab was performed in PBS?

Response: (Line 669-671 )The blocking solution was 1% bovine serum albumin (BSA)(Thermo Fisher Scientific, USA) in 1×PBST (Sangon Biotech, China) configuration. Primary and secondary antibodies were diluted separately with the blocking solution.

14Comment 3:For RFP detection lack of AbII species.

Response: (Line 677-678 )Goat Anti-Rabbit IgG H&L(HRP) secondary antibody

14Comment 4:Lack of rinsing steps between incubation with AbI and AbII and AbII and ECL.

Response: (Line 679-681 )After the blocking process, the two incubation processes were followed by three separate washes with 1×PBST,and the blocking and incubation processes were all stirred left and right on a shaker (Jarell, China).

15Comment: line 586 – if protein concentration was measured, and semi-quantitative WB performed – how many µg of crude protein were putted onto gel?

Response: The maximum amount of protein was 24.56 µg.

Minor comments:

16Comment: Line 35 – bacterial groups contains good and bad members… What is good and dad bacteria?

Response: (Line 37-38) human-unfriendly and human-friendly duality members.

17Comment: line 39 – please add more details of usage of Bcc complex in agriculture and medicine

Response: (Line 42-48)

Several strains are known to have biocontrol agents against phytopathogenic fungi, contribute to better water management, and improve nitrogen fifixation and plant growth[3].For instance, Burkholderia ambifaria and Burkholderia caribensis are presumably diazotrophic strains that promote growth of the grain crop amaranth[7]. EnacyloxinIIa from Burkholderia species was shown to have the most potent antibacterial activity against both Gram-positive and Gram-negative organisms[5].

18Comment: line 41 – natural products such as…

Response: (Line 49-54)

The discovery and exploitation of cloned, native recombinase genes enabled the activation of previously silent BGCs in Burkholderiales strain DSM7029, resulting in the isolation of glidopeptin[11].Examples include heterologous expression of BGCs encoding the lasso peptide capistruin79 and the polyketide−nonribosomal peptide glidobactin80 in E. coli[12].

19Comment: line 48 – cry218 encodes what

Response: (Line 61)cry218 gene, encoding the insecticidal crystal protein Cry1Ac

20Comment: line 67 – lack of reference

Response: (Line 81)[34,35]

21Comment: line 77  - in vivo italic

Response: (Line 89) in vivo

22Comment: line 106 – NA, MSA, MSA-ZN – lack of components, producer name and country

Response:

 (Line 397-400) the components of NA, MSA, MSA-ZN in materials and methods section

(Line 379-380) Sangon Biotech in China supplied the antibiotics and culture medium components (Sangon Biotech,China).

23Comment: line 116 – TP – lack of full name, producer name and country

Response: 

(Line 132)trimethoprim (Tp),

(Line 379-380)Sangon Biotech in China supplied the antibiotics and culture medium components (Sangon Biotech, China).

24Comment: line 119 – what is FPKM?

Response: (Line 145-146)fragments per kilobase of exon model per million mapped fragments (FPKM)

25Comment: line 159 – 153 – should be moved to Materials and Methods section

Response: (Line 576-580)Totally, the remaining 8 promoters were cloned into pBBR1-GMr-promoter-Luc-TPr using recombineering (Tables 1 and S1). Whether the B. pyrrocinia strain generated harbored these recombinants was first determined by plating on TP plates and then verified by PCR (Table S1). Transformants that obtained all eight promoters could be grown on TP plates.

26Comment: line 160 – what is RLU?

Response: (Line 189)the relative light unit (RLU)

27Comment: line 208 - what is GFP and RFP?

Response: (Line 247-248)

the green fluorescent protein (GFP), the red fluorescent protein (RFP)

28Comment: line 212 – Figure 6 (A-H)

Response: (Line 258-259 )

The scale is re-added.

We are very sorry, but due to a problem with the Zeiss fluorescence microscope in the laboratory, we were unable to successfully attempt a new experiment. We will include all of the original clearer brightfield images in the supplemental material.

29Comment: line 220 – A-H

Response: (Line 261)(A)-(H)

30Comment: line 221 – a - d

Response: (Line 262)(a)-(d),(e)-(h)

31Comment: line 356 – lack of software

Response: (Line 404-419)After sample extraction, the extracted total RNA was firstly removed from ribosomal RNA (rRNA) and then fragmented. Subsequently, double-stranded cDNA is synthesized. The double-stranded cDNA is then end-repaired, A-tailed and ligated to sequencing junctions. The ligated products are purified and amplified to obtain the final cDNA library. Finally, the constructed sequencing libraries are sequenced using the HiSeq sequencing platform. The sequenced data are called raw reads or raw data. The information is analyzed as fol-lows: first, the reads with low quality, contaminated junctions and high content of un-known bases N are removed from the raw data using the NGS QC Toolkit (2.3.3) to obtain clean reads. The clean reads were then aligned to the reference genome sequence using HISAT, and each sample was reconstructed using Rockhopper to obtain new transcripts. The well-known mRNA, novel mRNA were used as the reference gene set and the reads were compared to them using Bowtie2 software, and then the expression levels of the genes were calculated using RSEM software to calculate the expression levels of the genes. To perform the analysis of expression levels, sample correlation was done using the cor function in the R software. For linear correlation of the two data sets, pearson was chosen for the analysis.

32Comment: line 356 – lack of software producer name and country

Response: All software is provided by the Beijing Genomics Institute (BGI) and used to analyze.

33Comment: line 382 – lack of  polymerase producer name and country

Response: (Line 381 )(TransGen, China)

34Comment: line 458, 463- gene gfp and rfp encodes? Why Author choosed these two genes?

Response: (Line 247-248)The gene gfp encoding green fluorescent protein (GFP) and the gene rfp encoding red fluorescent protein (RFP), two reporter genes, which are simple and rapid tools for investigating promoter intensity.

35Comment: line 506, 508 – lack of  MSA producer name and country

Response: (Line 379-380)Sangon Biotech in China supplied the antibiotics and culture medium components (Sangon Biotech, China).

36Comment: line 507 – lack of  mini shaker producer name and country

Response: (Line 582) (RADOBIO, China)

37Comment: line 512 – lack of  centrifugation condition and temperature

Response: (Line 587-588)Cells were collected by centrifugation (Thermo, USA) at the speed of 12000 g for 2 min at 4°C and washed twice with PBS.

38Comment: line 512 –PBS with cations?  lack of producer name and country

Response: (Line 588) PBS(Vazyme,China)

Without cation, PBS buffer acts as a solvent to dissolve, adjust pH and protect the active substance.

39Comment: line 513 – OD600 was evaluated by using?

Response: (Line 589)

the cell density was adjusted (OD600=0.8) using NanoDrop 2000C (ThermoFisher Scientifc, USA)

40Comment: line 514 – lack of  reagents producer name and country

Response: (Line 380)(Sangon Biotech, China)

41Comment: line 516 – lysis buffer consist of?

Response: (Line 382-384)Firefly Luciferase Reporter Gene Aassy Kit (Yeasen, China), used to help luciferase emit bioluminomescence, contains cell lysis mixture and firefly luciferase assay reagent.

42Comment: line 517 – lack of  reagents producer name and country

Response: (Line 382)(Yeasen, China)

43Comment: line 531 – lack of  producer name

Response: (Line 609)(Applied Biosystems,USA)

44Comment: line 584 - lack of sonicator producer name and country

Response: (Line 664-665)individually sonicated by sonicator(Jingxin,China)

45Comment: line  - 587 – lack of % of the gel

Response: (Line 667)SDS-PAGE analysis of 12% gel

46Comment: line 588 – Authors used wet or dry transfer?

Response: wet transfer

47Comment: line 590 – lack of BSA producer name and country, diluent for BSA an information about agitation during blocking step and incubation with AbI. Lack of diluent for AbI

47Comment 1:lack of BSA producer name and country,

Response: (Line 670)(Thermo Fisher Scientific, USA)

47Comment 2:diluent for BSA an information about agitation during blocking step and incubation with AbI.

Response: (Line 669-682)The blocking solution was 1% bovine serum albumin (BSA)(Thermo Fisher Scientific, USA) in 1×PBST (Sangon Biotech, China) configuration. Primary and secondary antibodies were diluted separately with the blocking solution. Blots were blocked with 1% BSA for 1 h at room temperature, incubated with primary an-tibody of Anti-GFP antibody (1:10,000, Abcam, USA) overnight at 4°C, and then incubated with the secondary antibody of Goat Anti-Rabbit IgG (Alkaline Phosphatase) (1: 1000, Abcam, USA) in secondary antibody incubation solution and incubated for 1h at room temperature on a shaker. The primary antibody for Anti-RFP (1: 1000, Affinity, USA) was incubated overnight at 4°C, followed by incubation with Goat Anti-Rabbit IgG H&L(HRP) secondary antibody (1: 5000, Abcam, USA) for 1 h at room temperature. After the blocking process, the two incubation processes were followed by three separate washes with 1×PBST,and the blocking and incubation processes were all stirred left and right on a shaker (Jarell, China).

47Comment 3:Lack of diluent for AbI

Response: (Line 671)Primary and secondary antibodies were diluted separately with the blocking solution.

Moderate editing of English language.

Thank you for your suggestion, we will continue to improve the language in the later revisions.

We appreciate the constructive comments on key core issues and detailed suggestions on the finer points.

Reviewer 3 Report

Dear editor,  

In this manuscript (Optimization of Constitutive Promoters Using a Promoter Trapping Vector in Burkholderia pyrrocinia JK-SH007), the authors (Xue-Lian Wu and colleagues) designed a promoter trap system based on two reporter proteins adapted for promoter optimization in B. pyrrocinia strain JK-SH007 using firefly luciferase encoded by the luciferase gene set and trimethoprim (Tp)-resistant dihydro-20 folate reductase. My vote is to revise the manuscript before accepting it for publication. I have minor comments to improve the paper which is mentioned below.  

  • English language needs to be improved. The sentence construction and punctuation need to be better before the manuscript can be published. 

  • The subsection titles should be written properly to indicate what the authors want to conclude in these sections.  

  • Figures 1 ,2 and 3 can be combined into one figure as they are discussed in the same result subsection 2.1.  

  • Figures 4 and 5 can be merged into one figure. It would have been better if the values shown in figure 4 could be normalized to PPRL values. It was unclear to me if pCK values served as positive control or negative control. The Text in the section 2.2 is very confusing.  

  • Figures 6 and 7: They can be merged. The bright field images in figure 6 and the immunoblot images in figure 7 need to be replaced with better ones. I could not decipher the cells or bands properly. The molecular weight markers should be mentioned in the figure and the lanes should be properly marked.  

  • Table S2 can be moved to the main text as the promoter sequence information is essential to the manuscript.

Author Response

Responses to reviewers (original comments are in blue)

Reviewer #3:

In this manuscript (Optimization of Constitutive Promoters Using a Promoter Trapping Vector in Burkholderia pyrrocinia JK-SH007), the authors (Xue-Lian Wu and colleagues) designed a promoter trap system based on two reporter proteins adapted for promoter optimization in B. pyrrocinia strain JK-SH007 using firefly luciferase encoded by the luciferase gene set and trimethoprim (Tp)-resistant dihydro-20 folate reductase. My vote is to revise the manuscript before accepting it for publication. I have minor comments to improve the paper which is mentioned below.  

1 Comment: English language needs to be improved. The sentence construction and punctuation need to be better before the manuscript can be published. 

Response: Thank you for your suggestion, we will continue to improve the language in the later revisions, focusing on sentence structure and punctuation.

2 Comment: The subsection titles should be written properly to indicate what the authors want to conclude in these sections.  

Response: Thank you for pointing out the problems. We make the following changes:

(Line 225-246)Fluorescence detection in the engineered B. pyrrocinia JK-SH007 constructed by promoters

(Line 266-267)Western Blot analysis in the engineered B. pyrrocinia JK-SH007 constructed by promoters

(Line 615-616)Fluorescence detection in the engineered B. pyrrocinia JK-SH007 constructed by promoters

(Line 657-658)Western Blot analysis in the engineered B. pyrrocinia JK-SH007 constructed by promoters

3 Comment: Figures 1 ,2 and 3 can be combined into one figure as they are discussed in the same result subsection 2.1.

Response: (Line 125-126 )Special thanks to you for your good comments. Already merged into Figure 1.

4 Comment: Figures 4 and 5 can be merged into one figure. It would have been better if the values shown in figure 4 could be normalized to PPRL values. It was unclear to me if pCK values served as positive control or negative control. The Text in the section 2.2 is very confusing.  

4 Comment 1:Figures 4 and 5 can be merged into one figure. It would have been better if the values shown in figure 4 could be normalized to PPRL values.

Response: (Line 192-193)Thank you for your suggestion, I did not modify it because I was not familiar with the data normalization method.

4 Comment 2:It was unclear to me if pCK values served as positive control or negative control. The Text in the section 2.2 is very confusing.

Response: (Line 177-200 )We apologize for this inadvertently, misleading follow on word. The use of pCK to denote a negative control without a promoter is not appropriate. In the RLU assay, CK is the RLU standard and is used as a negative control to test for promoter facilitation. PRPL is the positive control. Reporter genes (Luc and TP r) without promoters served as negative controls (Figure S1), whereas those regulated by the inducible promoter PRPL served as positive controls.

5 Comment: Figures 6 and 7: They can be merged. The bright field images in figure 6 and the immunoblot images in figure 7 need to be replaced with better ones. I could not decipher the cells or bands properly. The molecular weight markers should be mentioned in the figure and the lanes should be properly marked.  

5 Comment 1: Figures 6 and 7: They can be merged.

Response: (Line 258-259,288-289)Thank you for the suggestion, we did not merge to ensure that the bacteria were visible under fluorescence and that the merged image was too large.

5 Comment 2:The bright field images in figure 6 and the immunoblot images in figure 7 need to be replaced with better ones. I could not decipher the cells or bands properly. The molecular weight markers should be mentioned in the figure and the lanes should be properly marked.

Response: We tried to put a large bright field image, but the bacteria were still not clear, so we put the original images in the supplementary material.

(Line 288-289)Figure 7 has been replaced.

6 Comment: Table S2 can be moved to the main text as the promoter sequence information is essential to the manuscript.

Response: (Line 166-167)Table S2 has been moved to Table 1.

We sincerely thank the reviewers for their comments on the article.

Round 2

Reviewer 1 Report

I am satisfied with the author's responses to my queries. 

English language is fine. Just minor grammar edits will be sufficient.